# Pre-adult aggression and its long-term behavioural consequences in crickets

**Julia S. Balsam, Paul A. Stevenson** *

Institute of Biology, Faculty of Life Sciences, Leipzig University, Leipzig, Germany

* stevenson@rz.uni-leipzig.de

**Data Availability Statement:** All relevant data are within the manuscript and its Supporting Information files.

**Funding:** This study was financed by a grant of the Deutsche Forschungsgemeinschaft (https://www.

## Abstract

Social experience, particularly aggression, is considered a major determinant of consistent inter-individual behavioural differences between animals of the same species and sex. We investigated the influence of pre-adult aggressive experience on future behaviour in male, last instar nymphs of the cricket *Gryllus bimaculatus*. We found that aggressive interactions between male nymphs are far less fierce than for adults in terms of duration and escalation. This appears to reflect immaturity of the sensory apparatus for releasing aggression, rather than the motor system controlling it. First, a comparison of the behavioural responses of nymphs and adults to mechanical antennal stimulation using freshly excised, untreated and hexane-washed antennae taken from nymphs and adults, indicate that nymphs neither respond to nor produce sex-specific cuticular semiochemicals important for releasing aggressive behaviour in adults. Second, treatment with the octopamine agonist chlordimeform could at least partially compensate for this deficit. In further contrast to adults, which become hyper-aggressive after victory, but submissive after defeat, such winner and loser effects are not apparent in nymphs. Aggressive competition between nymphs thus appears to have no consequence for future behaviour in crickets. Male nymphs are often attacked by adult males, but not by adult females. Furthermore, observations of nymphs raised in the presence, or absence of adult males, revealed that social subjugation by adult males leads to reduced aggressiveness and depressed exploratory behaviour when the nymphs become adult. We conclude that social subjugation by adults during pre-adult development of nymphs is a major determinant of consistent inter-individual behavioural differences in adult crickets.

## Introduction

Experiencing aggression, particularly social subjugation (social defeat stress), is recognised as a major factor inducing depression and depression-like symptoms in humans and animals [1] Accumulating evidence also points to links between aggression and consistent inter-individual behavioural differences, that had been described in many vertebrates and invertebrates, and is generally referred to as animal personality [2].

dfg.de/en/) to PAS, Grant Nr. STE 714/5-1. The
funders had no role in study design, data collection
and analysis, decision to publish, or preparation of
the manuscript.

**Competing interests:** The authors have declared
that no competing interests exist.

In this paper, we investigate the influence of pre-adult agonistic experience in male, last instar nymphs of the field cricket, *Gryllus bimaculatus*, which has advanced to a model system for studying the neurochemical control of aggression [3]. Pre-adult exposure to aggression is associated with various long-term, detrimental effects on behaviour in mammals including man [4,5] together with an increased risk for mental health problems [6]. There are, however, comparatively few studies of pre-adult aggression in invertebrates [7–9], and we know of only one study in crayfish that investigates its consequence for future adult behaviour [10].

In crickets, male nymphs are reported to exhibit "distinctive aggressive behaviours" that are "similar to those adopted by adults during aggressive disputes", whereby individuals that were "successful in competition monopolised access to food resources" [7]. This source also reports that "stress induced by increases in larval competition caused reductions in realised adult body size and decreased survival". The population density of nymphal crickets, which might also be expected to increase competition, was not, however, found to influence adult aggression or exploratory behaviour ("boldness", [11,12]). From studies to date it remains unclear, whether competition between nymphs establishes dominant-subordinate relationships as in adults, with potentially long lasting consequences for an individual´s future behaviour. As in most animals [13], adult male crickets that win an agonistic encounter are subsequently more aggressive [14], whereas losers become submissive [15–17]. Moreover, winners tend to have a more active behavioural profile than losers [18–20], which points to agonistic experience, in adults at least, as a prime determiner of consistent inter-individual behavioural differences in crickets [19,21].

The main aim of the current investigation is, therefore, to determine if pre-adult agonistic experience during nymphal life has any lasting effects, which could account for different behavioural profiles in adults. Our results show that aggressive behaviour is far less fierce in nymphs, and that their interactions do not result in winner or losers effects typical for adults. However, adult males aggressively intimidate male nymphs, and we provide evidence that this influences their future adult behavioural profile.

## Methods

### Experimental animals and ethical note

Experimental animals, Mediterranean field crickets, *Gryllus bimaculatus* (de Geer), were taken from a breeding colony maintained at the animal housing facility of the University of Leipzig since 25 years. Under our standard conditions, they were kept from hatching to the first nymphal stage until adult in groups of 20–30 in transparent plastic culture boxes (35 x 19 cm and 30 cm high), with sand covered floor and egg cartons for shelter at 22–24˚C, relative humidity 40–60%, 12 h: 12 h light: dark regime, with daily feeding on bran and fresh carrots and water *ad libitum*. All experiments were performed in the months of March to June and October to December, at room temperature (20–24˚C) during daylight hours, excluding midday and generally rainy, overcast days when aggression tends to be depressed [22,23]. All treatments conform to Principles of Laboratory Animal Care and German Law on Protection of Animals (*Deutsches Tierschutzgesetz*). A specific license for experiments on invertebrates is not required in Germany. All animals were returned to the breeding stock after the experiment. Our analysis is based on observations of 1032 crickets, of which none were used more than once for an experiment or again in other experiments unless explicitly stated otherwise.

For most experiments (Figs 1–5, and unless otherwise stated) we used male, last stage nymphs, and sexually mature adult males (>10 days after the final moult), that were taken from the standard breeding colony and kept socially isolated in individual glass jars for 48 h prior to experimentation, under the same ambient conditions with ample food and water

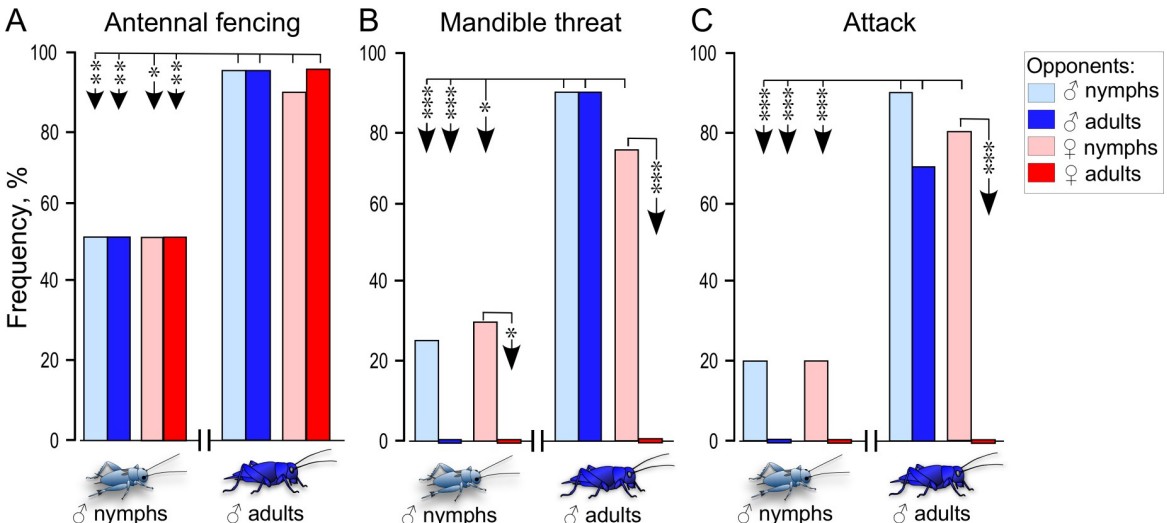

**Fig 1. Agonistic reactions of male nymphs and adults towards other nymphs and adults of both sexes.** Bars giving the frequency of (A) antennal fencing, (B) mandible threat display and (C) attack (lunging with physical engagement) of STI male nymphs and adults towards different opponents (see key in C): male nymphs (light blue bars), male adults (blue bars), female nymphs (light red bars) and female adults (red bars; N = 20 for each group, always comprising different animals). Significant differences between compared groups are indicated by arrows and given as asterisks (Fisher's exact tests: A * p < 0.05, ** p < 0.01; B and C * p < 0.025, *** p < 0.0005 after Bonferroni correction due to two comparisons).

("short term isolates", STI). On the day of the experiment, test-crickets was placed in a clear Perspex-glass rectangular observation arena (l. w. h.: 16 × 9 × 7 cm) with a sand-covered floor and left undisturbed for at least 5 min before commencing.

## Agonistic responses of nymphs and adults to each other's presence

For this experiment, we compared how individual male nymphs and adults responded to the presence of a single conspecific nymph or adult of both sexes (Fig 1). In each case, occurrences of the following elements of aggressive behaviour were observed and recorded: *antennal fencing*, in which two individuals lash each other's antennae (also shown at the beginning of courtship between adults); *mandible threat display*, when an individual spreads its mandibles in a characteristic threat display; *attack*, when one individual lunges and physically engages an opponent.

## Agonistic responses of nymphs and adults to manual antennal stimulation

Here we compared how often individual male nymphs and adults exhibited the mandible threat display in response to manual stimulation of their antennae (20 s) using the freshly severed antenna of various donor crickets that were anaesthetised by cooling (Fig 2). Donors were nymphs and adults of both sexes. In one test set, a mature adult male´s antenna was used that was first washed twice for 10 min with n-hexane (Sigma Aldrich, Deisenhofen, Germany) to remove cuticular pheromones [24,25]. In addition, we measured the angular extent of mandible spreading in response to antennal stimulation and during feeding (Fig 2). Spread angle was measured post event by analysing filmed responses (4K Video, Sony Alpha 6300 with Zeiss macro-zoom objective) on a computer (Dell Precision 3620, Round Rock, Texas, USA) with the software MB-Ruler (version 5.3, MB Softwaresolutions, Iffezheim, Germany).

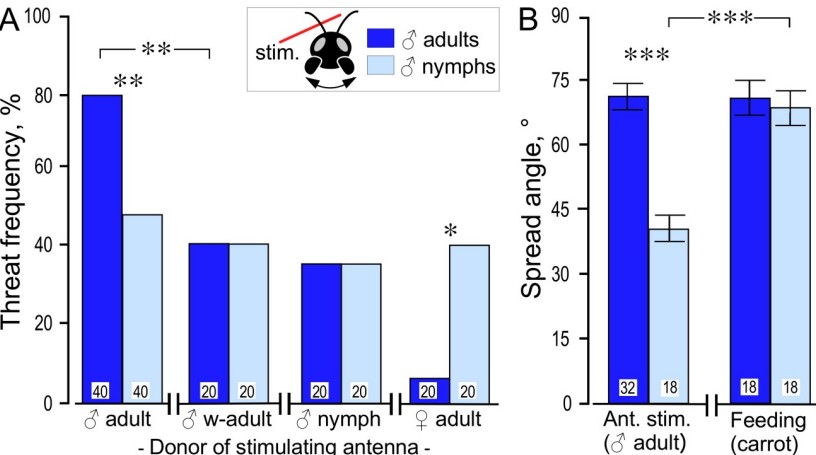

**Fig 2. Mandible threat display in response to antennal stimulation.** (A) Frequency of mandible threat for STI male adults and male nymphs (light and dark blue bars respectively) in response to mechanical stimulation with a freshly excised antenna from different donors (left to right): male adult, male adult and washed twice in n-hexane, male nymph and female adult. (B) Maximum angle of mandible spreading (mean and 95% CI) observed for male adults and nymphs in response to antennal stimulation (Ant. stim.) with a freshly excised male antenna and during feeding. Significant differences between data sets are indicated (A: Fisher's exact test, B: Student's unpaired t-test: * $p < 0.025$, ** $p < 0.005$, *** $p < 0.0005$; after Bonferroni correction due to two comparisons, N for each group is given in the bars).

## Evaluation of aggression

Aggressive behaviour was evaluated in dyadic contests between two equally sized males ($< 5\%$ weight difference) in the observation arena (Figs 3, 4, 5 and 6D). The opponents were placed at opposite ends of the arena and separated by a sliding door in the middle. The animals' interactions were then noted when the door was removed. As described elsewhere in detail, aggressive contests in crickets are characterised by a highly stereotyped series of increasingly aggressive actions, and can be scored on a simple scale (0–6) denoting the "level of aggression" to which the animals escalate [4,23]: *Level 0*: mutual avoidance (no aggression). *Level 1*: one cricket attacks, the other retreats (pre-established dominance). *Level 2*: antennal fencing. *Level 3*: mandible threat display by one cricket. *Level 4*: mandible threat displays by both crickets. *Level 5*: mandible engagement. *Level 6*: grappling (an all-out fight). Contest duration was measured to the nearest second as the time from initial contact up until establishment of a loser and winner with a stopwatch. On the few occasions the animals briefly lost contact to each other during a contest, the stopwatch was paused, so that only the duration of their interactions was recorded.

Since nymphs appeared to be far less aggressive than adults, we tested the influence of the following various treatments known to enhance aggression in adults in order to gain an indication of the maximal aggressive capacity of nymphs, and possible causes for it being less than in adults.

## Influence of winning and losing

In one set of experiments (Fig 3), pairs of adult males were first matched against each other to yield losers that retreat first and become submissive, and winners that generate the rival song and become hyper-aggressive [14,15]. Maximally 10 mins later, winners were re-matched against winners, and losers against losers. We attempted the same procedure for male nymphs. However, their interactions did not always generate clear winners and losers, so we only evaluated those interactions where this seemed apparent (see Results).

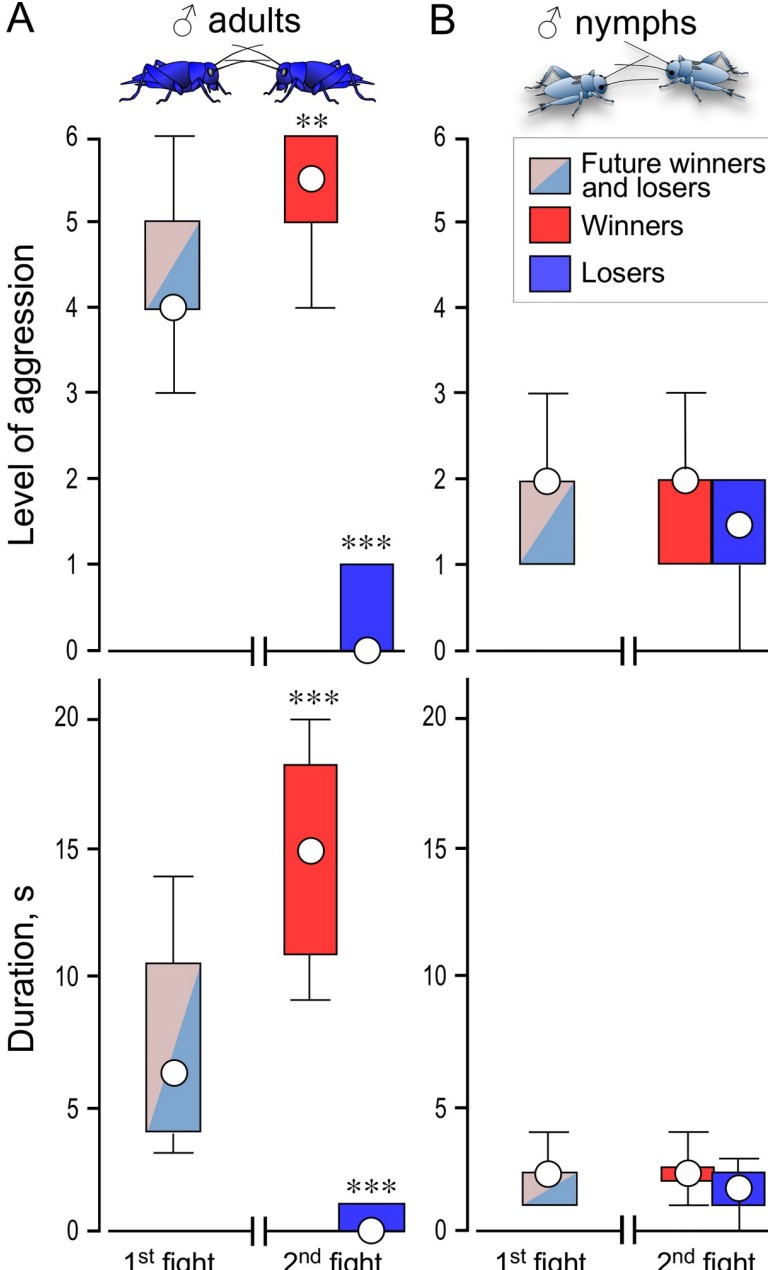

**Fig 3. Effects of aggressive experience on subsequent aggression.** (A) Male adults. (B) Male nymphs (both STI). Bars giving the level (top) and duration (bottom) of agonistic interactions between weight-matched pairs at their 1st fight after isolation (light red/light blue bars, future winners and future losers, N = 40), and later for a 2nd fight, staged between pairs of the resultant winners and pairs of resultant losers (red and blue bars respectively, N = 20 each). Circles: median, boxes: interquartile range, whiskers: 10th and 90th percentiles. Significant differences between the future and actual winners and future and actual losers are indicated (Wilcoxon signed-rank test: ** p < 0.01, *** p < 0.001). Note the absence of winner and loser effects in nymphs.

## Influence of prior antennal stimulation (*priming effect*)

Prior stimulation of an individual's antenna by manually stroking it with the freshly severed antenna of another adult donor male (priming) is known to increase aggression in adult males, particularly after losing a previous fight [26]. Here we compared how prior antennal

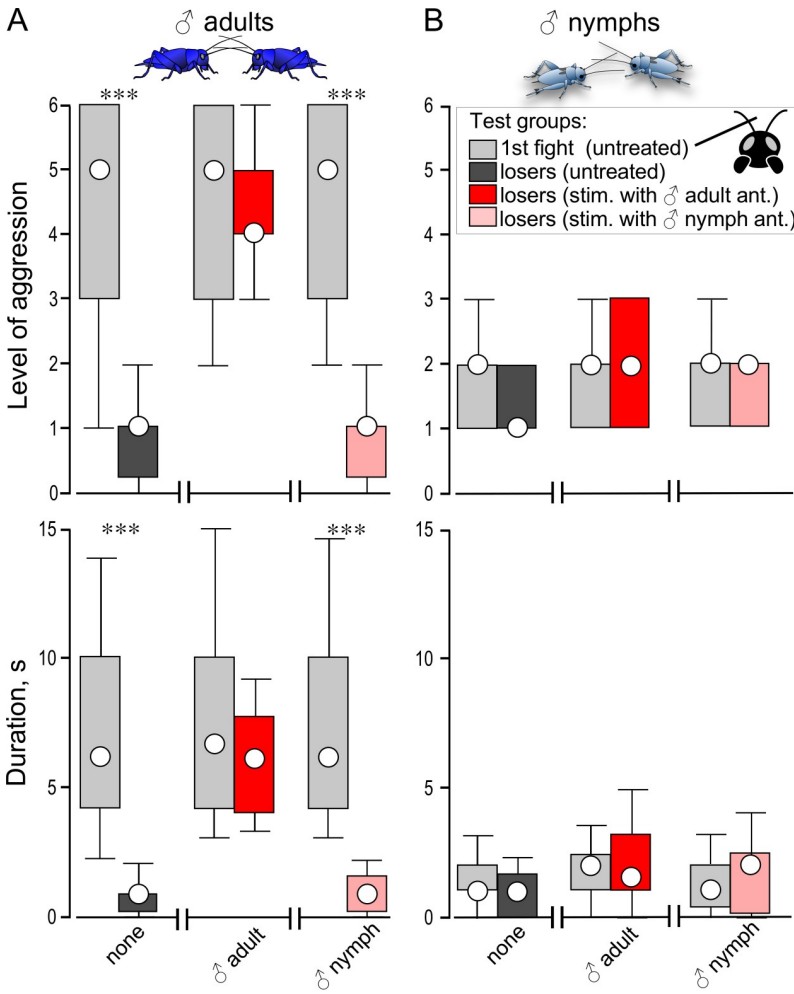

**Fig 4. Influence of prior antennal stimulation on subsequent aggression.** Bars giving the level (top) and duration (bottom) of agonistic interactions between weight matched pairs of STI male adults (A) and nymphs (B) at their 1st interaction (light grey bars) and for the same animals 10 min later. Before the second fight, the losers were either untreated (grey), or stimulated with an antenna from a male adult (red) or male nymph (light red). Circles: median, boxes: interquartile range, whiskers: 10th and 90th percentiles, N = 20 per group. Significant differences between paired data sets are indicated (Wilcoxon signed-rank test: *** p < 0.001).

stimulation influences aggression in adults and nymphs. We first staged contests between pairs of adults and between pairs of nymphs. The antenna of each resultant loser was then stimulated (for 20 s, 5 min post defeat) with the antennae donated from either a male adult or a male nymph. Each loser's aggressiveness was then evaluated 10 min later in a second contest against the previous winner (Fig 4).

## Influence of the octopamine agonist chlordimeform coupled with priming

Since the priming effect is increased dramatically in adults treated with the tissue permeable, irreversible octopamine agonist chlordimeform hydrochloride (CDM; [26]), we tested this procedure on male nymphs (Fig 5A). Individual nymphs were first anaesthetised by cooling and then injected with 20 μl of 1 mM CDM (Sigma Aldrich, Deisenhofen) in saline solution containing 1% DMSO (dimethylsulfoxide; saline components in mMol $L^{-1}$: NaCl 140, KCl 10,

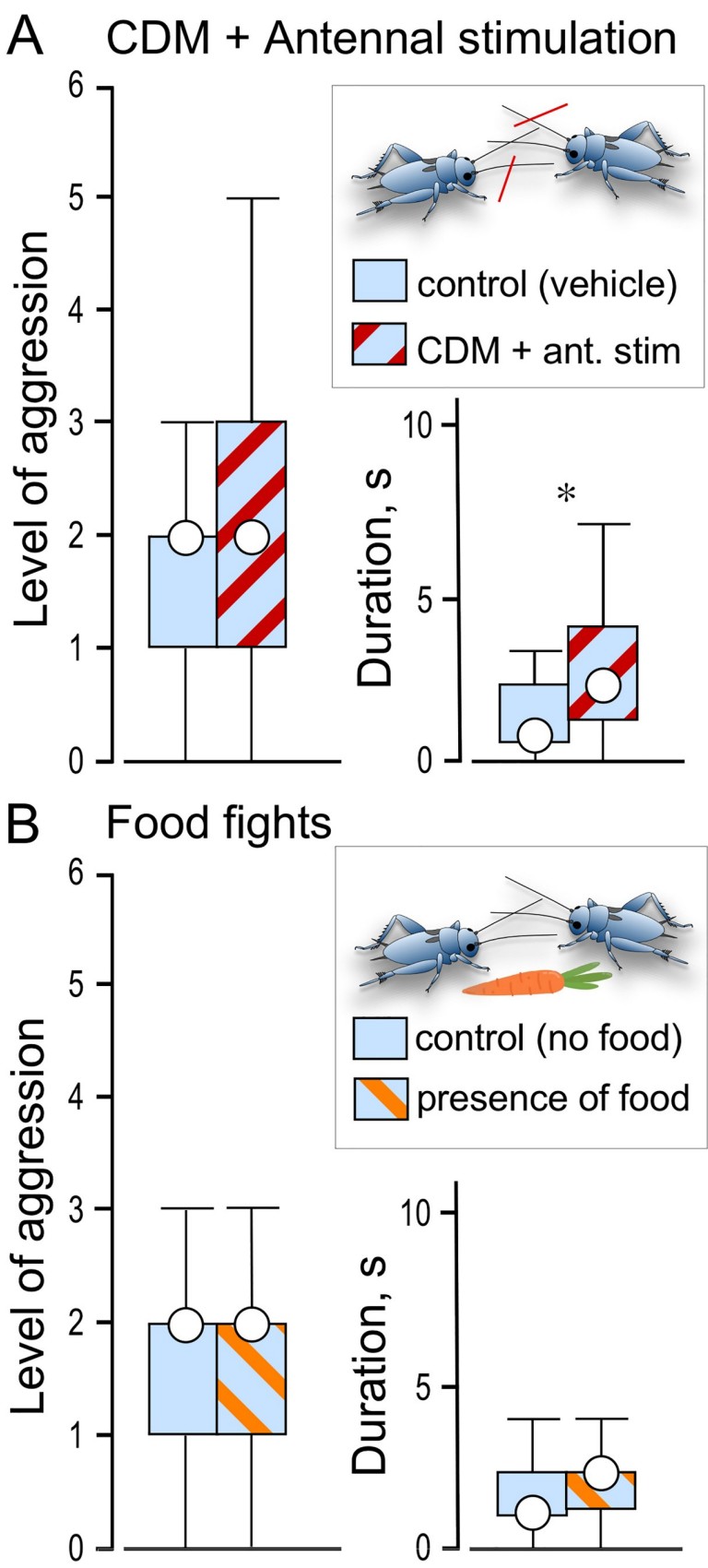

**Fig 5. Effects of aggression modulators on aggression in male nymphs.** Bars giving the level and duration of agonistic actions between STI male nymphs. (A) After treatment with the octopamine agonist chlordimeform (CDM in 1% DMSO, N = 13) combined with antennal stimulation (red hatched bars). (B) In the presence of food (orange hatched bars, N = 20). Controls (non-hatched bars) received DMSO in saline (A, N = 13) or no food (B, N = 20). Significant differences between unpaired data sets are indicated (Mann-Whitney *U* test: * $p < 0.05$).

$CaCl_2$ 7, $NaHCO_3$ 8, $MgCl_2$ 1, N-trismethyl-2-aminoethanesulfonic acid 5, D-trehalose dehydrate 4, pH 7.4) using a micro-syringe inserted through the membranous cuticle underlying the pronotal shield. Control animals received vehicle only (DMSO in saline). After waiting for 50–60 min, test crickets were subjected to antennal stimulation, as above, using an antenna from an adult male. Immediately afterwards, dyadic contests were staged between pairs of weight-matched treated nymphs.

## Influence of food as a resource

Since resources such as food [27] promotes aggression in adult crickets, we tested whether its presence also influences aggression in nymphs. For this, male nymphs were deprived of food for 48 h and their aggressiveness subsequently evaluated in dyadic contests staged in the immediate presence of freshly cut carrot (Fig 5B). Control contests were staged in the absence of food.

## Influence of social experiences in nymphs on their adult behaviour

To test for possible long-term behavioural effects of aggressive subjugation of nymphs by adults, we generated 3 cohorts of adult males housed under different social conditions (Fig 6A). One cohort was raised in the complete absence of adults as a group of 30–40 male nymphs from the moment of hatching until adult and sexually mature. A second cohort was raised in the same way, except that 10 sexually mature adult males were introduced into the nymphal colony as soon as the majority had moulted to the 4th nymphal stage. The third cohort differed only in that 10 sexually mature adult females were introduced into the nymphal colony.

Five to six days after introducing the adults, when the nymphs were between the 5th and 6th nymphal stage, we visually recorded the number of either nymph-nymph or adult-nymph attacks (lunges with physical engagement involving pushing and/or biting) in the 3 colonies for ten 1 h observation periods, spaced 2 per day (09:00–10:00 and 14:00–15:00), over 5 consecutive days. Other aspects of aggression could not be reliably evaluated under grouped conditions. Before each observation period, we first adjusted colony sizes to 30 nymphs (plus 10 adult males or females) by randomly removing nymphs if necessary. From the data gathered we calculated the mean number of attacks (and 95% CI) experienced per nymph per hour (Fig 6B).

Finally, individual nymphs were taken from the pure nymphal colony and the mixed nymph–adult male colony after moulting to the last nymphal stage, and kept socially isolated for at least 14 days until adult and fully sexually mature (long-term isolates, LTI). We then evaluated their general exploratory behaviour, as described below, and shortly after this their aggressiveness, as described above.

## Evaluation of exploratory behaviour

To compare the exploratory behaviour of LTI adult crickets raised either in the absence or presence of adult males, individuals were placed in the observation arena and their activity was

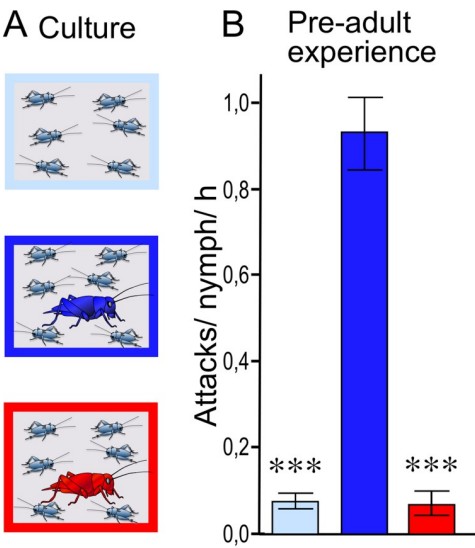

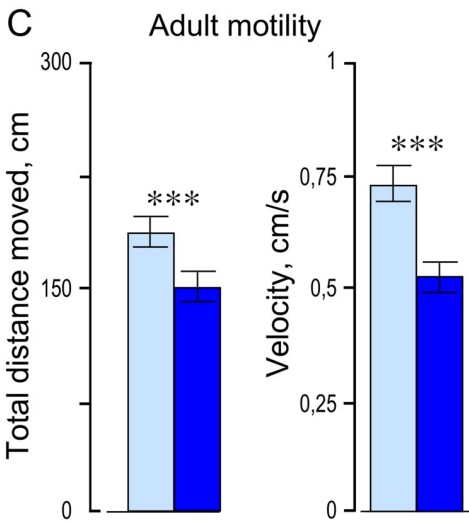

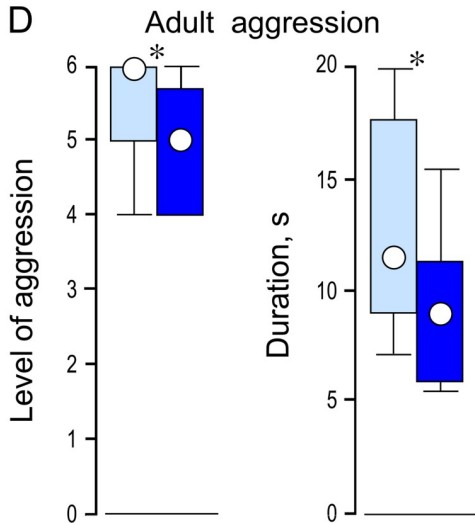

**Fig 6. Influence of social experiences in nymphs on their adult behaviour.** (A) Schema of culture conditions: 30 male nymphs were housed either without adults (light blue) from hatching until the last nymphal stage, or with 10 adult males (blue) or 10 adult females (red) from the 4th to the last nymphal stage. (B) Bars giving the mean (± 95% CI) number of attacks experienced per nymph per hour for 10 x 1 h sessions, from either other nymphs in the pure nymph culture (light blue, number divided by 3 to adjust for the larger number of nymphs in the cage), or adult males (dark blue) or adult females (red) in the mixed cultures. (C) Bars giving the distance moved (left) and mean velocity (right) of walking bouts during 5 min (means ± 95% CI, N = 20 per group) for LTI mature adult males taken >14 days previous as last stage nymphs from the pure nymph colony (light blue) and LTI adults from the mixed nymph-adult male colony (blue). (D) As for C, but giving the level (left) and duration (right) of fights between pairs of crickets from the two groups (circles: median, boxes: interquartile range, whiskers: 10th and 90th percentiles). Significant differences are indicated (B and C: Student's unpaired t-test: *** p < 0.0005 for B after Bonferroni correction due to two comparisons; p < 0.001 for C. D: Mann-Whitney U test: * p < 0.05).

recorded for 5 min with a digital video camera (Basler acA1920-155uc, Ahrensburg, Germany, 60 frames/s) and stored on a computer (Dell Precision 3620, Round Rock, Texas, USA) for subsequent analysis with video-tracking software (EthoVision XT14, Noldus, Wageningen, Netherlands) as given in detail elsewhere [19]. In brief, the centre point of each cricket was detected by the grey scaling method (minimum grey value 0, maximal 140) and its movement tracked on an X-Y coordinate system. To reduce noise from the animal's breathing movements, we set a "minimum distance moved" filter of 0.2 cm. For each animal we recorded the total distance moved as the length of the animals´ path covered in 5 min, and the mean walking velocity of bouts of actual walking, excluding times when the animal was stationary (Fig 6C).

## Statistical analysis

All statistical tests were performed using commercial software (Prism 7, GraphPad Software Inc., La Jolla, CA, USA) running on a Macintosh computer (Apple Computers, Cupertino, CA, USA). Fisher´s exact test was performed to compare relative frequencies of behavioural elements between groups. The Shapiro-Wilk test was used to test for data normality. For normally distributed data we give the means and 95% confidence interval (CI) of the mean. Student´s unpaired t-test was used to test for significant differences between the means of two independent groups. For nominal data (level of aggression) and fight duration, which was not normally distributed even after log transformation, we calculated the median and interquartile range (IQR) and applied non parametric procedures to test for significant differences in data distributions: the Mann-Whitney U test for unpaired data sets and the Wilcoxon signed-rank test for paired data sets. The numbers of crickets for each experiment and test group are indicated in the figures and/or legends. For single comparisons, the significance level alpha was set to p < 0.05. To compensate for errors due to multiple comparisons, we applied the Bonferroni correction to alpha for each time the same, or a mathematically related, set of numerical data was tested more than once (see reference [19] for reasoning and references). In the current study, there was never more than two comparisons (each is noted in text, alpha was set to < 0.025).

## Results

### Agonistic responses of nymphs and adults to each other's presence

On contact, male adults typically behave aggressively towards conspecific male adults, but not towards females which they court and try to copulate. Accordingly, as a first step towards evaluating the aggressiveness of male nymphs, we evaluated their behavioural responses on initial contact to other individual nymphs and adults of both sexes, and compared this with adult responses. Since antennal fencing is required to initiate fighting [26], and the mandible threat

display signals aggressive intent, while lunging with physical engagement ("attack") is a clearly aggressive action, we based this analysis on these three behaviours (Fig 1). As shown in Fig 1A, 50% of a total of 20 tested male nymphs showed antennal fencing behaviour towards conspecifics, regardless of their age group or sex. Male adults, in comparison, showed antennal fencing behaviour towards conspecifics, significantly more often and again indiscriminately of age-group or sex (90–95% Fisher's exact tests: p = 0.0033 for all comparisons, excepting responses towards female nymphs: p = 0.0138).

Male nymphs also showed the mandible threat display almost equally often towards both male nymphs (25%) and female nymphs (30%), but never towards male or female adults (0%; Fisher's exact tests for response towards adult females and males: p = 0.0202, 0.0471 respectively; latter insignificant after Bonferroni correction to alpha = 0.025). In comparison, male adults exhibited the mandible threat display significantly more frequently towards male nymphs (90%), male adults (90%) and female nymphs (75; Fisher's exact tests: p < 0.0005, < 0.0005, 0.0104 respectively), but never towards adult females (0%, significantly different to response to female nymphs, Fisher's exact test: p < 0.0005). Similarly, male nymphs occasionally attacked both male (20%) and female nymphs (20%), but never adult males and females (0%; differences not, however, significant: Fisher's exact tests: both p = 0.0305, with alpha set to 0.025). Adults, in comparison, attacked other adult males (70%), male nymphs (90%) and female nymphs (80%) significantly more often than male nymphs (Fisher's exact tests: p < 0.0005 for each case with alpha set to 0.025). Male adults, however, never attacked adult females (0%, significantly different to response to female nymphs, Fisher's exact test: p < 0.0005 with alpha set to 0.025).

## Agonistic responses of nymphs and adults to manual antennal stimulation

Our data so far indicate that cricket nymphs do not discriminate between male and female nymphs and that they also exhibit mandible threat displays less often than adult crickets on confronting a conspecific, particularly an adult male (Fig 1). To analyse this further, we evaluated the response of fresh cohorts of male nymphs and adults to manual stimulation of their antenna using the excised antennae taken from male and female adults and male nymphs as donors (Fig 2A). Confirming an earlier study on adults [26], adult male crickets frequently exhibited mandible threats in response to stroking their antenna with an adult male antenna (80%). Compared to this, male nymphs exhibited mandible threats significantly less frequently (45%, Fisher's exact test: p = 0.0024 with alpha set to 0.025). In response to stimulation with a hexane-washed male antenna, adult males showed the mandible threat response less often (40%, Fisher's exact test: p = 0.0034), but now equally often as male nymphs (40%). Similarly, male nymphs and adults showed an equal low incidence of mandible threats when stimulated with a male nymph's antenna (both 35%). Finally, in response to stimulation with an adult female antenna, male nymphs showed essentially the same frequency of mandible threat displays (40%), whereas adult males responded significantly less often (5%, Fisher's exact test: p = 0.0197).

As shown in Fig 2B, male nymphs also spread their mandibles significantly less wide in response to stimulation with an adult male antenna than adults (nymphs: mean 40˚, 95% CI: 36, 44, N = 18; adults: mean 71˚, 95% CI: 69, 74, N = 32; Student's unpaired t-test: p < 0.0005, alpha set to 0.025). To test if this reflects some physical inability, we also measured the mandible spread angle during feeding (Fig 2B, right). This revealed that nymphs open their mandibles significantly wider when feeding (mean 69˚, 95% CI: 66, 72, N = 18, Student's unpaired t-test: p < 0,0005, alpha set to 0.025), and now equally wide as adults (mean 72˚, 95% CI: 68, 75, N = 18).

## Influences of winning and losing

Since the experience of winning and losing an aggressive interaction has a strong effect on the expression of aggression in STI adult male crickets [3], we next tested how previous fighting experience influences subsequent aggressive behaviour in STI male nymphs. As control (Fig 3A), and confirming earlier studies [14,15], the first fights between STI adult male crickets typically escalated to level 4 (Median, IQR = 4, 4–5, N = 20) and lasted 6 s (Median, IQR = 6 s, 4–10.75, N = 20). Compared to this, contests between pairs of adult crickets that both won the previous encounter escalated to a significantly higher level of aggression and the fights lasted longer (level: Median, IQR = 5.5, 5–6, duration: Median, IQR = 15 s, 11–18, N = 20, Wilcoxon signed-rank tests compared to 1$^{st}$ fight: $p_{level}$ = 0.0023, $p_{duration}$ < 0.001). Conversely, contests between pairs of losers were significantly less fierce than at the first fight (level: Median, IQR = 0, 0–1, duration: Median, IQR = 0 s, 0–1, N = 20, Wilcoxon signed-rank tests compared to 1$^{st}$ fight: $p_{level}$ < 0.001, $p_{duration}$ < 0.001).

These winner and loser effects were not evident for dyadic contests between weight-matched STI male nymphs (Fig 3B). Because the winners and losers of a match between nymphs are not always immediately apparent, we only evaluated interactions in which the establishment of dominance seemed clear, in that a "loser" retreated from a "winner" that in turn continued to lunge at the opponent. Compared to adults, contests between male nymphs were less fierce in that they never escalated above level 3 (Median, IQR = 2, 1–2, N = 20) or lasted longer than 4 s (Median, IQR = 2 s, 1–2, N = 20). Furthermore, pairs of winners showed no significant change in their aggressiveness compared to the initial interaction (Wilcoxon signed-rank tests: $p_{level}$ = 0.8779, $p_{duration}$ = 0.05199). Similarly, interactions between losers were also not significantly different to those of the same animals at their first contest (Wilcoxon signed-rank tests: $p_{level}$ = 0.0586, $p_{duration}$ = 0.2825).

## Effect of prior antennal stimulation on fighting behaviour (*priming*)

We next tested whether nymphs show increased aggression after prior antennal stimulation as found previously for adults [26] and confirmed here. As shown in Fig 4A, STI adult losers of a first fight exhibit significantly reduced aggression when re-matched against their previous opponents at a second fight 10 mins later (Wilcoxon signed-rank tests: N = 20, $p_{level}$ < 0.001, $p_{duration}$ < 0.001). This loser effect is, however, no longer evident in losers after stimulating their antennae with the freshly excised antenna of another adult male (Wilcoxon signed-rank tests: N = 20, $p_{level}$ = 0.7327, $p_{duration}$ = 0.4479). Interestingly, we now found that this priming effect on adult loser aggression was not evident following mechanical stimulation with an antenna from a male nymph (Wilcoxon signed-rank tests: N = 20, $p_{level}$ < 0.001, $p_{duration}$ < 0.001).

Contrary to adults, mechanical antennal stimulation of male nymphs (Fig 4B) after losing an initial contest failed to have any significant effect on their aggression expressed at a subsequent, second contest, regardless of whether an adult male antenna (Wilcoxon signed-rank tests: N = 20, $p_{level}$ = 0.3398, $p_{duration}$ = 0.3029), or male nymph´s antenna (Wilcoxon signed-rank tests: N = 20, $p_{level}$ = 0.1289, $p_{duration}$ = 0.4469) was used as a stimulus.

## Priming and the octopamine receptor agonist chlordimeform (CDM)

In a further attempt to promote aggression in STI nymphs, we next evaluated their interactions after treatment with the octopamine receptor agonist chlordimeform (CDM) coupled with the aggression priming stimulus (antennal stimulation with an antenna from an adult male, Fig 5A). In STI adults, this leads to a dramatic increase in the escalation level and duration of subsequent aggressive encounters [26]. In nymphs, however, we observed no significant change in

escalation, but a just significant increase in contest duration (Mann-Whitney $U$ tests: N = 13, $p_{level} = 0.6267$, $p_{duration} = 0.0425$). Notably, however, one CDM-treated pair of nymphs showed actual physical fighting after prior antennal stimulation, during which the individuals interlocked their mandibles to push each other (level 5) in a contest that lasted 7 s, which is within the range typically exhibited by adults.

## Food fights

Earlier descriptions of competition in cricket nymphs were based in part on observing their interactions in the presence of food [7] which heightens aggression in adult crickets [27,28]. We tested whether the presence of food increased aspects of aggression in nymphs as measured by us (Fig 5B). This revealed no significant differences in the level of escalation or contest duration between pairs of STI nymphs that interacted in the absence or presence of food (Mann-Whitney $U$ tests: N = 20, $p_{level} = 0.9081$, $p_{duration} = 0.5490$).

## Influence of social experiences in nymphs on their adult behaviour

Since our data reveal that an adult male will attack a nymph on contact in the small fighting arena (Fig 1C), we next compared how often this occurs in colonies of 30 male nymphs raised either in the absence of adults, or together with 10 sexually mature adult males or females (Fig 6A). As shown in Fig 6B, nymphs raised without adults were rarely attacked by other nymphs under the grouped culture conditions (mean attacks endured per nymph per hour for n = 10 x 1 h sessions: 0.07, 95% CI: 0.06, 0.08). Similarly, male nymphs housed with adult females were also hardly ever attacked by them (mean attacks endured per nymph per hour: 0.07, 95% CI: 0.04, 0.09). Contrasting this, each individual nymph in the colony housed with adult males was attacked by an adult almost once an hour, and significantly more often than by other nymphs or adult females (mean attacks endured per nymph per hour: 0.93, 95% CI: 0.84, 1.02; Student's unpaired t-test: p < 0.0005 for both, with alpha set to 0.025).

With respect to general motility, video-tracking analysis revealed that adult males housed as nymphs without adult males were more active than those housed with adult males (Fig 6C). LTI adult male crickets that never had any social contact to adult males, walked a significantly greater total distance during a 5 min observation period (mean 189 cm, 95% CI: 174, 204, N = 20), compared to those subjected to potential subjugation by adult males as nymphs (mean 150 cm, 95% CI: 138, 162, N = 20; Student's unpaired t-test: p < 0.001). Furthermore, the mean velocity of the walking episodes was significantly faster for adult males from the pure nymph colony (mean 0.74 cm/s, 95% CI: 0.67, 0.81, N = 20), compared to adult males from the mixed nymphal-adult colony (mean 0.54 cm/s 95% CI: 0.49, 0.59, N = 20; Student's unpaired t-test: p < 0.001)

Finally, an evaluation of aggression revealed that LTI adults from the pure nymph colony were significantly more aggressive than LTI from the mixed nymph-adult colony (Fig 6D, Mann-Whitney $U$ tests: N = 20, $p_{level} = 0.0337$, $p_{duration} = 0.0223$).

## Discussion

Agonistic experience, and in particular social defeat, is becoming recognised as a major factor that induces symptoms of severe depression and shapes consistent inter-individual differences in behaviour (animal personality) in numerous animal groups [1,2]. In this paper we analysed the consequences of pre-adult (nymphal) agonistic experiences in crickets, *Gryllus bimaculatus*, which have become a model system for investigating the neurochemical control of aggression [3]. Earlier reports imply that aggression exhibited between male, last instar cricket nymphs is essentially the same as that between adult males, and has consequences for adult

fitness, at least in terms of body size and survival rate [7]. Our study, however, which compares aggression in nymphs and adults, with a focus on how it influences future behaviour, calls for a different interpretation.

We show that agonistic interactions between weight-matched male nymphs are far less fierce than those between mature adult males and nymph-nymph interactions are typified by a significantly lower level of aggressive escalation and duration (Fig 3) Actual physical fights were, under normal circumstances, never observed (see however below). Our observations are not essentially different to those of earlier investigators, who noted that nymphs escalate less [20] and do not show grappling and flipping behaviour typical for adults [29], which would correspond to our level 5 and 6 fights. Of more relevance to the present study competition between nymphs does not result in winner and loser effects (Fig 3B), which in adults influences the intensity of subsequent aggressive interactions and the establishment of dominant-subordinate relationships with additional consequences for the expression of other behaviours [19].

These findings raise the question why nymphs do not show the all-out fighting characteristic for mature adult males? Most studies of hemimetabolous insects, including locusts and crickets, indicate that the central nervous apparatus underlying adult-specific behaviours such as flight, copulation and phonotaxis are already functional in nymphs [30–33], but cannot be normally recruited by the natural releasing stimulus [30]. We propose that the same applies to aggression. Firstly, and as also noted by Simmons [7], male nymphs that contact each other clearly exhibit some aspects of adult aggressive behaviour, including antennal fencing, mandible threat displays and lunging, but far less often (Fig 1). Furthermore, even though nymphs do not spread their mandibles as wide as adults on confronting another male, they are fully able to do so, at least during feeding (Fig 2B). Reduced mandible spreading in nymphs is also unlikely to be a cause of reduced aggressiveness, since neither mandible quality [34] nor the ability to use them [35], has any influence on contest intensity or duration. Hence, cricket nymphs appear to be as capable as adults of exhibiting actual fighting behaviour, they only seem less inclined to do so.

Since social isolation is generally associated with increased aggression, it might be argued that nymphs could become more aggressive, if they were isolated for longer than 48 h as in our study. However, in adult crickets at least, there is no difference in aggression between individuals isolated for 1, 2 or 6 days [15]. Furthermore, low aggression in grouped and briefly (< 1 h) isolated adult crickets has been shown to result from prior social subjugation, and that isolation merely allows recovery from the loser-effect to a default aggressive state [15]. Since nymphal, agonistic interactions seem to have no consequence for their future agonistic behaviour (Fig 3B), it appears unlikely that their aggressiveness would increase with longer social isolation.

The most parsimonious explanation for the limited aggressiveness of nymphs is that their ability to respond to the natural aggression-releasing stimulus is not fully developed. In adult crickets, antennal fencing behaviour between males on first contact involves mechanical stimulation with transfer of male-specific semiochemicals that are essential for initiating normal aggression [24–26,36–39]. Notably, and contrasting the central nervous system [40], the sensory system of hemimetabolous insects undergoes dramatic changes during post-embryonic development, including the strengthening of synaptic contacts of existing afferents and the birth of numerous new ones with each moult [41]. Such changes in olfactory receptor neurones during development in locusts are known to result in increased chemo-afferent convergence on the antennal lobe with maturation [42]. Our experiments suggest a corresponding scenario in crickets. In contrast to adults, nymphs do not discriminate between male and female nymphs since they show aggressive actions such as mandible threats and attacks almost equally towards both (Fig 1). Adult males also show these same elements of aggression on

contacting both male and female nymphs, but practically never on contacting an adult female. Similarly, with respect to antennal stimulation with another antenna, most adult males show mandible spreading when the donor was an adult male, about half when a male nymph, and hardly none when an adult female. Male nymphs in contrast do not discriminate between the age and sex of the donor antenna (Fig 2A). Furthermore, adult losers become more aggressive after stimulation with an adult male antenna, but not when the antenna is from a male nymph (Fig 4). Together, these data suggest that cricket nymphs neither produce nor respond fully to adult contact pheromones. While differences in pheromonal signatures between the sexes are clearly documented in adult crickets (*Teleogryllus ocanicus*: [38]), studies of insect pheromones in nymphs are rare (e.g. [43]) and to our knowledge none focus on pheromones controlling social behaviours, such as aggression. There is only evidence for developmental changes in the quality and quantity of cuticular hydrocarbons during adulthood [24].

Overt aggression in adult insects may also depend on developmental changes in the octopaminergic system. In crickets, adult male-male contact leads to release of octopamine [36] and to direct excitation of octopamine containing neurones by specific pheromone sensitive receptors in *Drosophila* [44]. While octopamine is not essential for the expression [23] or initiation [26] of aggression, it increases the propensity to escalate and persist longer in fighting by raising an individual's threshold to flee [3]. Thus, a wide variety of experiences that promote aggression, including prior physical exertion (flying, fighting), antennal contact with another adult male's antenna, burrow residency, the presence of an adult female or food and winning a previous encounter, are each mediated by the neuromodulatory action of octopamine [45,46]. Contrary to adults, however, neither winning (Fig 3), stimulation with an adult male's antenna (Fig 4), nor the presence of food (Fig 5) led to any significant increase in aggression expressed by nymphs. This difference to adults could be due to immaturity in the pheromone-linked octopamine system. For example, adult crickets (female *Acheta domesticus*, males not tested) contain 50% more octopamine in their brains than last instar nymphs [47] and octopaminergic stimulation of the second messenger adenylate cyclase is more potent in the brain of adult locusts compared to nymphs [48]. Supporting this octopamine deficiency idea, treatment with the octopamine agonist chlordimeform (CDM) significantly increased the duration of contests between nymphs (Fig 5A). On one occasion, the treated nymphs even exhibited mandible engagement (level 5) in a contest that lasted several seconds, which we otherwise only encountered in adults. This observed exception supports our proposal that the central motor circuitry underlying male aggressive behaviour is in principle fully functional in the last stage cricket nymphs. In conclusion, we propose that the reduced aggressiveness of nymphs reflects developmental immaturity in both the pheromonal and octopaminergic systems required for initiating and maintaining aggressive behaviour in adults.

A main focus of this study was to determine the extent to which agonistic experience in nymphs influences their future behaviour. We found no evidence that agonistic competition between last stage male nymphs had any lasting behavioural consequences. Their interactions do not always generate clear winners and losers, and even when this was reasonably clear, winners were not subsequently more aggressive, nor were the losers less so. Nymphs are, however, aggressively intimidated by adult males (Fig 1), on average about once an hour in grouped nymph-adult colonies (Fig 6B), and this has long lasting behavioural consequences. In adult crickets, repeated hourly defeats induce long-term depression of aggression that can last for days [16,17]. Similarly, nymphs raised with adult males are significantly less active and less aggressive weeks later when they become adult, in comparison to adults with no contact to adult males during nymphal life (Fig 6C and 6D). In adults, the depressing effect of chronic social defeat is mediated by nitric oxide and serotonin [17, 49], so that we are currently addressing the role of these neuromodulators in mediating experience dependant modulation

of behavioural profiles during development. We conclude that social subjugation by adult males during pre-adult life is a major factor forging consistent, life-long, inter-individual behavioural differences between adults (animal personality).

## Supporting information

**S1 File.**
(ZIP)

## Acknowledgments

This manuscript forms part of the doctoral thesis of JSB. We thank Jan Rillich, numerous members of Robert Kittel's group at Leipzig and the 4 referees for constructive critic and helpful comments on our manuscript.

## Author Contributions

**Conceptualization:** Julia S. Balsam, Paul A. Stevenson.

**Data curation:** Julia S. Balsam.

**Formal analysis:** Julia S. Balsam, Paul A. Stevenson.

**Funding acquisition:** Paul A. Stevenson.

**Investigation:** Julia S. Balsam.

**Methodology:** Paul A. Stevenson.

**Project administration:** Paul A. Stevenson.

**Resources:** Paul A. Stevenson.

**Supervision:** Paul A. Stevenson.

**Writing – original draft:** Julia S. Balsam, Paul A. Stevenson.

**Writing – review & editing:** Julia S. Balsam, Paul A. Stevenson.

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
