## [Decision Letter · Decision Letter 0]

27 Jan 2020

PONE-D-19-35154

Pre-adult aggression and its long-term behavioural consequences in crickets

PLOS ONE

Dear Prof. Dr. Stevenson,

Thank you for submitting your manuscript to PLOS ONE. After careful consideration, your work is suitable for publication, pending minor revisions. Therefore, we invite you to submit a revised version of the manuscript that addresses the points raised during the review process.

We would appreciate receiving your revised manuscript by Mar 12 2020 11:59PM. To enhance the reproducibility of your results, we recommend that if applicable you deposit your laboratory protocols in protocols.io, where a protocol can be assigned its own identifier (DOI) such that it can be cited independently in the future. For instructions see: http://journals.plos.org/plosone/s/submission-guidelines#loc-laboratory-protocols

We look forward to receiving your revised manuscript.

Kind regards,

Sergio D. Iñiguez, Ph.D.

Academic Editor

PLOS ONE

Journal Requirements:

Reviewers' comments:

Reviewer's Responses to Questions

**Comments to the Author**

1. Is the manuscript technically sound, and do the data support the conclusions?

Reviewer #1: Yes

Reviewer #2: Yes

Reviewer #3: Yes

Reviewer #4: Yes

2. Has the statistical analysis been performed appropriately and rigorously? 

Reviewer #1: Yes

Reviewer #2: No

Reviewer #3: Yes

Reviewer #4: Yes

3. Have the authors made all data underlying the findings in their manuscript fully available?

Reviewer #1: Yes

Reviewer #2: Yes

Reviewer #3: Yes

Reviewer #4: Yes

4. Is the manuscript presented in an intelligible fashion and written in standard English?

Reviewer #1: Yes

Reviewer #2: Yes

Reviewer #3: Yes

Reviewer #4: No

5. Review Comments to the Author

Reviewer #1: The authors present a creative approach to the study of social experience (aggression) as a major determinant of behavior differences between animals of the same species “animal personality”. The authors build upon an established model system using field cricket, Gryllus bimaculatus, for studying the neurochemical control of aggression. This is quite relevant as it is known that pre-adult exposure to aggression is associated with various long-term, detrimental effects on behavior in mammals including man together with an increased risk for mental health problems. Specifically, the study addresses not only pre-adult aggressive behavior while adding the consequences for their future behavior as adults.

The methods for the study are elaborate and intricate and makes use of a 25 year old colony. Fig 1 and 2 examine behavioral responses to conspecific contact and antennal stimulation. The results here are categorized in and recorded as: antennal fencing, in which two individuals lash each other’s antennae; individual opens its mandibles; attack, when one individual lunges and physically engages an opponent; mandible spread angle: the angular extent of mandible spreading. It is not clear in the presentation of the results as to why these battery of behaviors are the most adequate. Fig 3-6 examine aggressive behavior and was evaluated in dyadic contests between two equally sized males (< 5% weight difference) in the observation arena. Mechanistically, it is unclear how the experiments on the influence of prior antennal stimulation (priming effect) were done and how these results are necessary in addition to the priming of aggression (this sounds like a positive control) ? Statistical Analyses are adequate for the data. Overall, the main finding of this study, reveal that agonistic interactions between weight-matched male nymphs are less fierce than those between mature adult males. Nymph-nymph interactions are typified by a significantly lower level of aggressive escalation and duration. Authors suggest that these decrease is explained by the possibility that nymph's ability to generate and respond to the natural aggression-releasing stimulus is

not fully developed. For instance, changes in olfactory receptor neurons during development in locusts are known to result in increased chemo-afferent convergence on the antennal lobe with maturation. This difference to adults could reflect developmental immaturity in both the pheromonal and octopamine systems

Reviewer #2: Authors investigated the influence of agonistic encounter of juvenile male crickets on future behaviour. Authors found aggressive interactions between juveniles are few less escalation to establish dominant-subordinate relationship and also no winner/loser effects are distinguished . The topic is interesting and their methods are reasonable. Parts of the results are potentially useful to understand agonistic interactions of arthropod animals. I do however have some major queries which need to be clarified.

Major concerns:

1) Winner and loser effects are usually defined as follows: A previously winning experience increases the winning probability of the next agonistic encounter, whereas a previous losing experience has the opposite effect. Authors did not show in this study that the winners increased their winning probability and the losers decreased it. Authors indicated only increase or decrease of aggression level of fight-experienced crickets. Is there any direct evidences to show correlation between aggression level and agonistic outcomes, since Mustafa wt al. (2019) reported that increase in aggressive level has no advantage of agonistic bouts in zebrafish (Behav Brain Res 270:111942).

2) Authors indicated that aggressive behaviour is far less fierce in juveniles and that their interactions do not result in winner or losers effects typical for adults. I agree authors' conclusion according to the results presented in this paper. However, I am very afraid that a rather short period of isolation (48 h) leads to these results. Simmons (1987; ref 7) and Abe et al. (2018; ref 17) have reported that juvenile male crickets exhibit distinguish aggressive behaviours and form discrete dominant-subordinate relationship. In their experiments, juvenile animals are isolated much longer period after final moult of juveniles. In actual, aggressive level of losing crickets increased to 5 after long-term isolation of more than 14 days in this paper (Fig. 6D) that is higher than that of naive adult males (median 4) isolated 48 h (Fig. 3A). Thus, the results might be changed if juveniles isolated more longer are used in this study. Please argue this possibility to Discussion section.

3) As my understanding, two different statistical analysis were performed in Figure 1 and also Figure 2. First analyses is that the occurrence frequency of a particular behavioural act (e.g. antennal fencing) to the same opponent was compared between juvenile and adult groups. And the second analyses is that the difference of juvenile (or adult) responses to 4 opponent groups were compared. Argues of this difference of comparison of groups are rather ambiguous to understand. Authors must clarify this point. If authors want to compare the difference of adult male's response to 4 different opponents simultaneously, significance level alpha must be set 0.05/(3+2+1) = 0.0083. I could not understand meaning of L234-237. I think summarized table(s) of statistical analysis would be helpful to understand the treatments (or groups) of multiple comparisons.

Minor comments:

L44: Delete However

L184: when adult means when they became adult?

L203-204: meaning is unclear.

L249-250 and many places: significance level alpha must be set 0.0083.

L276: meaning "due to two comparisons" is unclear. Authors compared 4 groups.

L287-288: delete with alpha set to 0.025

L289: against who?

L421: Contrasting these?

L424: alpha set to 0.017 (= 0.05/3)

Reviewer #3: This is an interesting study looking and analysing agonistic behaviour in larval crickets, for which only few data exists. It has been performed in an expert lab, most familiar with the experimental procedures involved. The data are well measured and clearly presented, but see comment below.

It is interesting to see the differences in aggression behaviour between larval and adult crickets. The outcome that subjugation of larval crickets may have a lasting effect on their behaviour as adults is very interesting.

The text reads a bit wordy and repetitive at times and should be streamlined. In a similar way the discussion is a bit repetitive, i.e. results are repeated in detail instead of summarised. Also some key words like “personality” are rather thrown into the text in brackets. I think the MS would gain by briefly/critically explaining these key words a bit more, without over emphasizing.

Regarding the figures: Some of the distributions in Figs 3 to 5 look identical, which comes as a surprise, please comment. May be overlaying the original data points with the bar charts could make the data representation more clear.

I have some further minor notes, given in the annotated PDF.

Reviewer #4: Summary: In this study, the authors examine how agonistic interactions of last instar Gryllus bimaculatus nymphs can influence the expression of their adult behavior. Though the agonistic behavior of adult crickets has been the focus of numerous prior studies, the behavior of cricket nymphs has received much less attention, and the authors indicate that no previous studies have examined the impact of nymph social experience on their subsequent adult agonistic behavior. Here, the authors report that cricket nymphs are much less aggressive than adults, as has been previously reported by others. However, the authors expand on this by thoroughly examining whether various factors already known to increase aggression in adult crickets (ex. social status, hunger/presence of food, etc.) could also alter the aggressiveness of male nymph. They report that none could. The authors also examined how a nymph’s prior social experience with adult crickets can impact its own adult behavior. They found that male nymphs raised with adult males were less motile and less aggressive upon reaching adulthood than were male nymphs that were just raised with other nymphs.

Critique:

This study addresses an interesting question that has not received much attention in the past. The study was carefully conducted and the manuscript is well organized. The methods provide sufficient detail. The figures are well composed and easily understood, and appropriate statistical analyses were used. The authors provide a focused discussion that effectively puts their findings in context and the abstract is a good overall summary of the key points of the study. Though errors are not excessive, this reviewer does recommend further editing of the entire manuscript for language and grammar. Some suggested revisions: 1. The introduction is a concise overview of the topic. However, the authors should provide references for their statement of “widely reported” agonistic behavior of pre-adult invertebrates (line 63). 2. The direct quotes taken from reference #7 in the introduction (lines 65-67) should be removed; instead, this information should be summarized in the authors’ own words. 3. In methods, a clearer explanation of how pauses in activity were ‘deducted’ (line 141) is needed. 4. The units of ‘20µl/1 mM CDA’ used by the authors should instead be ‘20 µl of a 1 mM CDA’ (line 167). 5. The arrows in Fig 1 should be explained in the legend. 6. The authors data show that nymphs do not open their mandibles as wide as adults during antennal stimulation (Fig. 2B), but the opposite is stated in the text (line 304). 7. ‘Proactive’ (ex. line 426) should be replaced with the word ‘active’ throughout the manuscript. 8. Fig. 6: It would have been interesting to see the adult data for male nymphs raised with adult females. This data could have further supported the idea that it was specifically the aggressive attacks by the adult males that were key to these findings. Alternatively, it could have answered the question of whether simply interacting with adults of any sex could influence a nymph’s adult aggressiveness. However, the results do clearly support the importance of a nymph’s social history can impact its future behavior.

6. PLOS authors have the option to publish the peer review history of their article (what does this mean?). If published, this will include your full peer review and any attached files.

Reviewer #1: Yes: Ulises M. Ricoy

Reviewer #2: No

Reviewer #3: No

Reviewer #4: No

---

## [Author Response · Author response to Decision Letter 0]

12 Feb 2020

Response to Reviewers (PONE-D-19-35154)

Reviewer 1

1) Fig 1 and 2 examine behavioral responses to conspecific contact and antennal stimulation. The results here are categorized in and recorded as: antennal fencing, in which two individuals lash each other’s antennae; individual opens its mandibles; attack, when one individual lunges and physically engages an opponent; mandible spread angle: the angular extent of mandible spreading. It is not clear in the presentation of the results as to why this battery of behaviors are the most adequate. 

We have changed the heading of the first section of Results to: “Agonistic responses of nymphs and adults to each other’s presence” and added the following in results: “Since antennal fencing is required to initiate fighting [26], and the mandible threat display signals aggressive intent, while lunging with physical engagement (“attack”) is a clearly aggressive action, we based this analysis on these three behaviours (Fig 1).

We have also added the following in Method (Evaluation of aggression): “As described in detail elsewhere [4,20], aggressive contests in crickets are characterised by a highly stereotyped series of increasingly aggressive actions, and can be scored on a simple scale (0 – 6) denoting the “level of aggression” to which the animals escalate.”

2) Mechanistically, it is unclear how the experiments on the influence of prior antennal stimulation (priming effect) were done and how these results are necessary in addition to the priming of aggression (this sounds like a positive control)? 

As also suggested by another referee, we have amended the Methods (Influence of prior antennal stimulation) to make our procedure more clear: "We first staged contests between pairs of adults and between pairs of nymphs. The antennae of each resultant loser was then stimulated (20 s, 5 min post defeat) with the antennae donated from either a male adult or a male nymph. Each loser’s aggressiveness was then evaluated 10 min later in a second contest against the previous winner (Fig 4).” Please note, that the priming experiments were performed to see if prior antennal stimulation increases aggression in nymphs, as shown for adults. To clarify this, we now state in Methods: “Since nymphs appeared to be far less aggressive than adults, we tested the influence of the following various treatments known to enhance aggression in adults in order to gain an indication of the maximal aggressive capacity of nymphs, and possible causes for it being less than in adults.

3) Authors suggest that these decrease is explained by the possibility that nymph's ability to generate and respond to the natural aggression-releasing stimulus is not fully developed. .…. This difference to adults could reflect developmental immaturity in both the pheromonal and octopamine systems.

We thank the referee for his positive comments here. To make our arguments even clearer we have rearranged the section in the discussion, and conclude as suggested with: “In conclusion, we propose that the reduced aggressiveness of nymphs reflects developmental immaturity in both the pheromonal and octopaminergic systems required for initiating and maintaining aggressive behaviour in adults”.

Reviewer 2

1) Winner and loser effects are usually defined as follows: A previously winning experience increases the winning probability of the next agonistic encounter, whereas a previous losing experience has the opposite effect. Authors did not show in this study that the winners increased their winning probability and the losers decreased it. Authors indicated only increase or decrease of aggression level of fight-experienced crickets. Is there any direct evidences to show correlation between aggression level and agonistic outcomes, since Mustafa wt al. (2019) reported that increase in aggressive level has no advantage of agonistic bouts in zebrafish (Behav Brain Res 270:111942).

Please note that we also show that winning increased fight duration, and it is known that animals that persist longer are the ones that win (we have observed this in crickets many times). Persistence is also a more sensitive measure, particular so for nymphs where the winners are not always clearly apparent (as mentioned also in the results).

2) Authors indicated that aggressive behaviour is far less fierce in juveniles and that their interactions do not result in winner or losers effects typical for adults. I agree authors' conclusion according to the results presented in this paper. However, I am very afraid that a rather short period of isolation (48 h) leads to these results. Simmons (1987; ref 7) and Abe et al. (2018; ref 17) have reported that juvenile male crickets exhibit distinguish aggressive behaviours and form discrete dominant-subordinate relationship. In their experiments, juvenile animals are isolated much longer period after final moult of juveniles. In actual, aggressive level of losing crickets increased to 5 after long-term isolation of more than 14 days in this paper (Fig. 6D) that is higher than that of naive adult males (median 4) isolated 48 h (Fig. 3A). Thus, the results might be changed if juveniles isolated more longer are used in this study. Please argue this possibility to Discussion section.

To accommodate this critic, we now state in the discussion: “Since social isolation is generally associated with increased aggression, it might be argued that nymphs could become more aggressive if they were isolated for longer than 48 h as in our study. However, in adult crickets at least, there is no difference in aggression between individuals isolated for 1, 2 or 6 days [15]. Furthermore, low aggression in grouped and briefly (< 1 h) isolated adult crickets has been shown to result from prior social subjugation, and that isolation merely allows recovery from the loser-effect to a default aggressive states [15]. Since nymphal, agonistic interactions appear to have no consequence for their future agonstic behaviour (Fig 3B), it appears unlikely that their aggressiveness would increase with longer social isolation.

Regarding Fig 6D, the observation that LTI show higher aggression than short term isolates (STI Fig 3A) is something reported in an earlier paper, and is not surprising since LTIs have had practically no previous adverse social agonistic experience as adults (cited from reference 17). 

Regarding isolation period, the earlier authors write “Sixth instar larvae (were) placed individually into small Petri dishes” (ref. 7) or that crickets were isolated “after juveniles moulted to the final instar stage” (ref. 20), but we found no clear statement when exactly their aggression was evaluated, but this is now beside the point we think.

3) As my understanding, two different statistical analyses were performed in Figure 1 and also Figure 2. First analyses is that the occurrence frequency of a particular behavioural act (e.g. antennal fencing) to the same opponent was compared between juvenile and adult groups. And the second analyses is that the difference of juvenile (or adult) responses to 4 opponent groups were compared. Argues of this difference of comparison of groups are rather ambiguous to understand. Authors must clarify this point. If authors want to compare the difference of adult male's response to 4 different opponents simultaneously, significance level alpha must be set 0.05/(3+2+1) = 0.0083. I could not understand meaning of L234-237. I think summarized table(s) of statistical analysis would be helpful to understand the treatments (or groups) of multiple comparisons.

Sorry, it seems that our original text to the experiments in Figs 1 and 2 have misled the referee to assume this data is from the same experiment (or is a repeated experiment) using the same animals, which is not the case. To avoid this confusion, we now give the methods for these experiments in 2 separate sections, and have changed the section heading leading to Fig 1 in the text to: “Agonistic responses of nymphs and adults to each other’s presence” and the text section heading leading to Fig 2 to “Agonistic responses of nymphs and adults to manual antennal stimulation”. 

To clarify that the data in Fig 2 (and all other Figs) are based on fresh observations of different individuals, we now state in Methods: “Our analysis is based on observations of 1032 crickets, of which none were used more than once for an experiment or again in other experiments unless explicitly stated otherwise.” 

To avoid any misunderstandings regarding multiple comparisons, we now provide the following text in Methods (Statistical analysis): “For single comparisons, the significance level alpha was set to p < 0.05. To compensate for errors due to multiple comparisons, we applied the Bonferroni correction to alpha for each time the same, or a mathematically related, set of numerical data was tested more than once. In the current study, there was never more than two comparisons (alpha is noted in text or legends). 

We would also like to point out, that here is no consensus on when data sets should be included as part of an entire test family when adjusting test statistics for multiple comparisons (for arguments and references see [19]). Our standpoint, now given in the methods, has been accepted e.g. by the journal Animal Behaviour (Elsevier), which is particularly strict regarding multiple comparisons. 

Minor points 

L44: Delete However - DONE

L184: when adult means when they became adult? Yes, Rephrased

L203-204: meaning is unclear. DONE REWORDED

L249-250 and many places: significance level alpha must be set 0.0083. NO WE COMPARE 2 GROUPS (SEE POINT 3 ABOVE)

L276: meaning "due to two comparisons" is unclear. Authors compared 4 groups. NO WE COMPARE 2 GROUPS (SEE POINT 3 ABOVE)

L287-288: delete with alpha set to 0.025 TEXT NOW CHANGE (SEE POINT 3 ABOVE)

L289: against who? IN RESPONSE TO STIMULATION WITH A HEXANE-WASHED ANTENNA

L421: Contrasting these? NO CHANGE (THIS IS CORRECT HERE WE THINK)

L424: alpha set to 0.017 (= 0.05/3) NO WE COMPARE 2 GROUPS (SEE POINT 3 ABOVE)

Reviewer 3

1) The text reads a bit wordy and repetitive at times and should be streamlined. In a similar way the discussion is a bit repetitive, i.e. results are repeated in detail instead of summarised. Also some key words like “personality” are rather thrown into the text in brackets. I think the MS would gain by briefly/critically explaining these key words a bit more, without over emphasizing.

We thank this reviewer for offering numerous very helpful suggestions for improvement in the marked PDF. We have adopted all the changed indicated. We have also shortened the indicated sections in the discussion and deleted repetitions.

Regarding the term personality: We suspect that this reviewer may find this term inappropriate, as we do ourselves, but we would prefer to keep back any critical comments at present. However, to comply with the request, we now use the term animal personality without quotes and change an introductory sentence to read: Experiencing aggression, particularly social subjugation (social defeat stress), is recognised as a major factor inducing depression and depression-like symptoms in humans and animals [1]. Accumulating evidence also points to links between aggression and consistent inter-individual behavioural differences, described in recent years in many vertebrates and invertebrates, and currently referred to as animal personality [2]. 

2) Regarding the figures: Some of the distributions in Figs 3 to 5 look identical, which comes as a surprise, please comment. May be overlaying the original data points with the bar charts could make the data representation more clear.

Our account was not clear here, and has been changed. For example, the grey bars in Fig 3 have been changed to a single bar, it depicts the data at the first fight between “future” winners and losers (light red/light blue bar) - hence the scores are the same. We only separated the group so that we could make a paired comparison to the same animals at the second fight – which we still do, but show only one bar. We hope the new legend makes this clear. 

Some of bars in figs 4 and 5 also look similar, particularly for losers. Here the data are from different animals sets - their performances are however very consistent. We have tried putting all data points on the graph, but this is detrimental to clarity in the graphs.

Reviewer 4

1) The authors should provide references for their statement of “widely reported” agonistic behavior of pre-adult invertebrates. 

A very good point! We have now changed the text to read: “There are comparatively few studies of pre-adult aggression in invertebrates [7-9]”, and we know of only one study in crayfish that investigates its consequence for future behaviour when adult [10].” Please note: references 8-10 are new.

2) The direct quotes taken from reference #7 in the introduction (lines 65-67) should be removed; instead, this information should be summarized in the authors’ own words. 

We plea to leave the quotes, it is our explicit intent to cite the author exactly, as we do not wish to give out interpretation of the author’s statements. 

3) In methods, a clearer explanation of how pauses in activity were ‘deducted’ (line 141) is needed. 

DONE. We now write: “On the few occasions that the animals briefly lost contact to each other during a contest, the stopwatch was paused, so that only the duration of their interactions was recorded.” 

4) The units of ‘20µl/1 mM CDA’ used by the authors should instead be ‘20 µl of a 1 mM CDA’

DONE - changed as requested

5) The arrows in Fig 1 should be explained in the legend. DONE

6) The authors data show that nymphs do not open their mandibles as wide as adults during antennal stimulation (Fig. 2B), but the opposite is stated in the text (line 304). 

DONE - text amended

7) ‘Proactive’ (ex. line 426) should be replaced with the word ‘active’ throughout the manuscript. 

DONE

8) It would have been interesting to see the adult data for male nymphs raised with adult females. This data could have further supported the idea that it was specifically the aggressive attacks by the adult males that were key to these findings. Alternatively, it could have answered the question of whether simply interacting with adults of any sex could influence a nymph’s adult aggressiveness. However, the results do clearly support the importance of a nymph’s social history can impact its future behavior.

In retrospect, yes this would have been a good addition. But we would not expect any effect since the female do no intimidate the nymphs (see Fig 6B).

---

## [Decision Letter · Decision Letter 1]

9 Mar 2020

Pre-adult aggression and its long-term behavioural consequences in crickets

PONE-D-19-35154R1

Dear Dr. Stevenson,

We are pleased to inform you that your manuscript has been judged scientifically suitable for publication and will be formally accepted for publication once it complies with all outstanding technical requirements.

With kind regards,

Sergio D. Iñiguez, Ph.D.

Academic Editor

PLOS ONE
---

## [Editor Report · Acceptance letter]

12 Mar 2020

PONE-D-19-35154R1 

Pre-adult aggression and its long-term behavioural consequences in crickets 

Dear Dr. Stevenson:

I am pleased to inform you that your manuscript has been deemed suitable for publication in PLOS ONE. Congratulations! Your manuscript is now with our production department. 

With kind regards,

on behalf of

Dr. Sergio D. Iñiguez 

Academic Editor

PLOS ONE